# Evolutionary game analysis of supply chain financing for film and television enterprises considering co-production

Ye Zhen *, Hui Li, Wenli Wang

School of Economics and Management, Taiyuan University of Science and Technology, Taiyuan, China

* zhenye@tyust.edu.cn

## Abstract

Currently, many film and television enterprises struggle with insufficient funds during the visualization process. To address this issue, companies can opt to enter into co-production agreements with sales platforms, implement supply chain financing, obtain financing through banks, or collaborate with all three parties simultaneously. The paper presents a tripartite evolutionary game model involving film and television enterprises, sales platforms, and banks and discusses the strategic choices of operation and the investment and financing behaviors of the three parties. Additionally, it analyzes film and television enterprises' capital, film production costs, copyright acquisition costs, and other factors and examines the impact of these factors on their strategic choices. The study shows that only when film and television enterprises have strong negotiation abilities will a sales platform consider establishing a cooperative relationship with them. As the cost of film production increases, the requirement for the negotiation ability of film and television companies also increases accordingly. Once the cooperation of the sales platform is obtained, film and television enterprises can actively complete the film shooting. If the sales platform chooses not to cooperate, the film and television enterprises' capital will be restricted. In the case of non-cooperation, if the film production cost is low, enterprises can still choose to complete the film using bank financing. In contrast, even if they have obtained the support of bank loans, the high production cost of the film compels the film and television enterprises to adopt the strategy of negative completion. The research provides theoretical support for further alleviating the financing difficulties of film and television enterprises and at the same time provides a theoretical basis for the cooperation among film and television enterprises, sales platforms, and banks.

## 1. Introduction

Since 2012, China's film industry has entered a golden era characterized by unprecedented historical opportunities and prosperity (Qiushi Journal, 2020). Guided and

**Data availability statement:** All relevant data are within the paper and its Supporting information files.

**Funding:** This work is supported by the Youth Fund for Humanities and Social Sciences Research of the Ministry of Education [Grant number: 21YJC630172] and the Project of National Natural Science Foundation of China [Grant number: 72571189, 72171162].

**Competing interests:** NO authors have competing interests Enter: The authors have declared that no competing interests exist.

supported by national policies, the film and television industry in China has embarked on a new epoch; however, financing challenges have persistently hindered the high-quality development of film and television enterprises [1,2]. As a major factor of production, capital plays an extremely important role in both the creative production stage and the release and screening stage of film and television works [3]. However, due to factors such as limited scale, difficult quantification of intangible assets and uncertain expected returns, many investment institutions adopt a wait-and-see attitude towards film and television enterprises and find it difficult to provide financial support [4,5].

As a traditional financing channel, banks play a major role in the financing process of film and television enterprises. As a core value of film and television products, copyright serves as an important element through which film and television enterprises pledge financing to banks. In 2007, the Beijing branch of the Bank of Communications and the Beijing Sky Star Film and Television Culture Communication Company signed a loan contract with copyright as a pledge for the TV series "Lotus Lamp"; this case received the first copyright pledge loan in China's film and television industry. The Law of the People's Republic of China on the Promotion of the Film Industry promulgated by the State in 2016 proposes that the state encourages financial institutions to carry out film-related intellectual property pledge financing business in accordance with the law and supports the development of the film industry through credit and other means. Subsequently, the bank copyright pledge loan model was further promoted. Since 2018, the Bank of Shanghai has established a cooperative relationship with the crew of the Opening year drama, "Flowers." This bank provided financing and capital needs analysis for enterprises and recommended suitable financial service products, which bring mutually beneficial opportunities for both parties. Xu et al. (2019) discussed the intellectual property pledge financing cooperation mechanism between commercial banks and technology-based SMEs with the participation of third-party intermediary platforms, pointing out that, for film and television enterprises with less capital, copyright financing is in line with the development of the film and television industry, and the copyright of film and television works is used as collateral to obtain financing support. The process is conducive to the increase of film and television enterprises' profits [6–8].

Previous studies mostly focused on the relationship between film and television enterprises and banks or between film and television enterprises and platforms and failed to build a game model based on the three parties of film and television enterprises, sales platforms, and banks. With the continuous development of game theory, multi-party evolutionary game theory has attracted much attention as a new research hotspot. This theory analyzes the real problems by simulating the dynamic mechanism of biological evolution, and provides a new way to solve the complex game problems. Compared with traditional game theory, evolutionary game pays more attention to the stability of players' strategies. Its formation and development went through a long period of time, and it was not applied to the scope of economics until 1980. In the financing process of film and television enterprises, they may face different market environments and the choice of other game players' competitive

strategies. The use of evolutionary game theory can better analyze how film and television enterprises choose the optimal strategy in the process of evolution, and how these strategies adjust with the changes of other game players' strategies, help us better understand the cooperation and competition between these players, and thus provide a better plan for cooperation and coordination between them. Based on this, we provides a new solution for the financing of film and television enterprises by constructing a supply chain financing mode of tripartite cooperation among film and television enterprises, sales platform and banks, and from the perspective of game theory.

In our mode, film and television enterprises raise money from banks to produce works and sell them through sales platforms, forming an efficient and safe financing method. We analyzes the key factors that affect the behavior choice of different strategic agents and better understands the conflict of interest and cooperation mechanism between the parties. It addresses the following questions: Under which conditions will the sales platform choose to cooperate with film and television enterprises? Will film and television companies choose to actively complete the shooting process? What factors affect banks' willingness to lend? The present research provides a certain theoretical basis for the cooperation between film and television enterprises, sales platforms, and banks in real life and promotes the high-quality development of film and television enterprises. The contribution of this paper lies in revealing the influence mechanism of the cooperation between film and television enterprises and sales platforms on the lending behavior of banks, as well as the effects of factors such as negotiation ability, the cost of copyright acquisition and the proportion of equity in joint production on dynamic evolution, providing suggestions for the selection of the optimal cooperation model for film and television enterprises. We use the evolutionary game model to reveal the learning behaviors of irrational decision-makers and the dynamic characteristics of the equilibrium strategies.

## 2. Literature review

Although many countries have introduced policies in recent years to encourage innovation and the development of financial products suitable for film and television culture enterprises, some countries also give discount interest rate support to high-quality film and television culture enterprises. However, the special content and asset attributes of the film and television industry motivates most financial institutions to give priority to large enterprises to avoid risks, thus the financing difficulties of small- and medium-sized film and television enterprises have not been solved. In response to this problem, some scholars have also explored other financing models, such as Goettler and Leslie [9], who found that film and television enterprises prefer to adopt the co-financing model. This model is more effective in solving the problem of big-budget films in the film and television industry. Thomson [10] pointed out that American film and television enterprises have various financing methods, including loan financing, pre-sale agreement financing, and publisher financing. Mina and Hasnan [11] studied the crowdfunding financing model of film and television enterprises and analyzed the influence of decision making of film producers, supporters, distributors, and platform owners. Zhan [8] asserted that cooperation among the government, film and television enterprises, and banks should be strengthened. In addition, the cooperation between film and television enterprises and platforms has also become a research hotspot. Pu et al. [12] revealed that, after the platform obtains high-quality content, it can coordinate market access strategies with film and television enterprises to jointly promote the successful promotion of works. The cooperation between the two parties enables the network platform to provide diversified high-quality content to attract more users to subscribe and watch and maximize profits while reducing investment risks. At the same time, film and television enterprises can make use of the extensive coverage and promotion ability of network platforms to spread their works to more audiences [13–15]. At present, many popular dramas have been produced as a result of the joint action of film and television market forces, such as film and television companies and sales platforms. For example, the hit dramas "Chang 'an twelve hours" and "Dear, love" are products of film and television enterprises and sales platforms. Zeng [16] pointed out that the rise of video-sharing platforms provided a broader stage for the promotion and publicity of film and television works and promoted the prosperity and diversified development of the film and television works industry. Vany [17] reveals that for small and medium-sized film and television enterprises, the

high-risk nature of the film and television industry lies in the excessively concentrated use of funds and the large uncertainty of income. Under the background of unbalanced and inadequate development of the regional film and television industry, Xie [18] pointed out that the source of financing difficulties in the film industry for a long time was that film and television enterprises could not provide corresponding guarantees, so they could not obtain loans. Wang (2014) starting from the uncertainty of film investment, pointed out that in the current financing structure of Chinese films, traditional bank loans and non-professional investors are still the main. By analyzing the expectation right of film copyright, Huang and Chen [19] intuitively and comprehensively analyzed and summarized the current situation and predicament of copyright pledge in China's film industry, further indicating that copyright is the core value of film and television products, and copyright financing is an important form of intangible asset financing for film and television enterprises. Zhen et al. [20] pointed out that copyright pledge financing can make full use of the value of copyright assets of enterprises and effectively alleviate the financing difficulties of enterprises. Jusufi [21] made an in-depth study of the crowdfunding financing model and explained that the difference in financing methods has an important impact on the development of enterprises. Zhu [22] analyzed and compared the crowdfunded film model with the traditional film model to explore the existing risk research, indicating that film and television enterprises can solve the problem of financial constraints through crowdfunding.

Although different financing methods address the urgent needs of film and television enterprises to a certain extent, in actual operation, due to the different funds and resources invested by different entities and the inconsistent rights and interests of all parties, differences and games often arise in the aspects of profit distribution, copyright authorization, financing mode, and contract constraints [23]. In the digital era, the control rights of film and television copyright owners have been significantly weakened, and many copyright infringements have emerged on video-sharing platforms. Researchers have asserted the benefit that would arise if the two jointly built an integrated film industry ecosystem [24,25]. In the process of financing, it is necessary to combine the characteristics of the film and television industry, apply modern enterprise financing theory, and fully explore the innovation potential of the capital market, so as to better solve the financing difficulties of film and television enterprises [26]. However, most of the existing literature uses qualitative methods such as policy and case studies to analyze the factors influencing the financing of film and television enterprises and the countermeasures to alleviate the problem of financing difficulties for small and medium-sized enterprises; there are also studies using empirical research methods to analyze the factors influencing the success of crowdfunding for film and television enterprises and the paths for film and television enterprises to implement crowdfunding financing. The existing research methods are limited to the decision-making of individual enterprises and cannot study the strategic interactions among supply chain member enterprises and banks. Therefore, this paper uses the game theory to study the cooperative behavior between film and television enterprises and the platform, as well as the lending behavior of banks, which break through the limitations of the one-way causal relationship in traditional decision-making theories. Furthermore, the main advantage of the evolutionary game model lies in its dynamic adaptability. It simulates the real process in which decision-makers adjust their strategies through learning.

## 3. Model assumptions and construction

### 3.1. Model assumptions

Consider the three-party evolutionary game model composed of film and television enterprises, sales platforms, and banks. Due to market uncertainty, information asymmetry, and other factors, the decision makers in the model struggle to make the optimal decision in one game and need to adjust to develop a stable strategy, as the decision makers all possess finite rationality [Xie, 2002; Wang and Cai (2024); 27–30]. Film and television enterprises, sales platforms, and banks each have two different strategies to choose, and film and television enterprises in the filming process can choose (positive completion or negative completion). Positive completion indicates that film and television enterprises

need to invest in more costs and the production scale is larger. It is assumed that the film and television enterprises choose the probability of positive completion is $x(0 \leq x \leq 1)$. In negative completion, when the film and television enterprise input costs are reduced, the scale of production is reduced, and the probability is $1 - x$. The sales platform can choose (cooperation or no cooperation) when it receives a co-production application from the film and television enterprise. The cooperation of the sales platform is regarded as the co-production of the two enterprises with the probability of $y(0 \leq y \leq 1)$, while non-cooperation is regarded as a one-time payment of the acquisition cost of the movie screening rights by the platform after the completion of the shooting of the work, with the probability of $1 - y$. When banks receive an application for financing from film and television enterprises, they can choose (loan or no loan). The probability that the bank chooses to make loans is $z(0 \leq z \leq 1)$, and the probability that it does not make a loan is $1 - z$. The other main relevant assumptions of the model are as follows:

*Assumption 1*: When the sales platform chooses to cooperate with the film and television enterprises, if the bank lends money, the film and television enterprises and the sales platform have the same ratio of revenue sharing and cost sharing, and the ratio between the two sides is $a, 1 - a$. If the bank doesn't lend money due to the limited capital of the film and television enterprises, the two sides cooperate to create a profit-sharing plan according to the proportion of capital contribution, and the distribution ratio is $F/c, \; 1 - F/c$. This setting is consistent with equity allocation in practice. When the sales platform does not cooperate, after the filming is completed, the sales platform has to purchase the screening rights of the film at a cost of $g$ according to the negotiating ability of the film and television enterprises. The stronger the negotiation ability of the film and television enterprises, the higher the cost of copyright acquisition. This setting is consistent with most operations research literature about wholesale price contract [31,32].

*Assumption 2*: The film and television enterprises have their own funds, $F$, which are lower than the cost of completing the film $c$. The amount of financing that the film and television enterprises obtain from the bank is variable and affected by the strategic choices of the sales platforms as well as their own strategic choices. This assumption is consistent with most operations research literature about supply chain finance [28,33,34].

*Assumption 3*: If the sales platform cooperates with the film and television enterprises during the bank financing period, the bank does not need to manage the financing amount. When the sales platform does not cooperate, the bank needs to pay $k$ proportion of the management fee. This assumption is consistent with Wang and Cai (2014), who use the fixed bank fee as the administration and related costs.

*Assumption 4*: The revenue $p$ obtained when the filming of the film is completed is greater than the principal and interest sum of all filming costs of the film and television enterprises' loans from the bank, i.e., $p > c(1 + r)$.

### 3.2. Model construction

According to the assumptions above, Table 1 summarizes the relevant parameters and meanings in this paper.

As a result, this paper can derive the benefits of film and television enterprises, sales platforms, and banks under different strategy choices. As shown in Table 2. Each combination in the revenue matrix from top to bottom is the profit of film and television enterprises, sales platforms, and banks, respectively.

## 4. Model analysis

### 4.1. Strategic stability analysis of film and television enterprises

By using the revenue matrix in Table 2, this paper concludes that the expected revenue of film and television enterprises at positive completion is:

$$E_{x1} = yz(p-c)(a - \frac{F}{c}) - zr(c-F)(1-y) + y\frac{F}{c}(p-c) + z(1-y)(g-c) - yzr(ac-F) \tag{1}$$

**Table 1. Explanation of symbols.**

| Variables | Explanation of symbols |
|---|---|
| $a$ | The ratio of revenue and cost sharing |
| $p$ | Proceeds from the completion of the movie |
| $c$ | Costs incurred by film and television enterprises for completion of filming |
| $F$ | Own funds of film and television enterprises |
| $q$ | Scale of cost and revenue reduction in negative completion of film and television enterprises ($0 < q < 1$) |
| $g$ | Copyright acquisition cost |
| $r$ | Bank loan interest rate |
| $k$ | Proportion of administrative costs paid by banks providing financing |

**Table 2. Revenue matrix for film and television enterprises, sales platforms, and banks.**

| Strategies Selection | Sales Platform Cooperation $y$ | | Sales Platform No-cooperation $1-y$ | |
|---|---|---|---|---|
| | Bank Loan $z$ | Bank No-loan $1-z$ | Bank Loan $z$ | Bank No-loan $1-z$ |
| Film and television enterprises | $a(p-c)-(ac-F)r$ | $\frac{F}{c}(p-c)$ | $g-c-(c-F)r$ | 0 |
| Positive Completion | $(1-a)(p-c)$ | $(1-\frac{F}{c})(p-c)$ | $p-g$ | 0 |
| $x$ | $(ac-F)r$ | 0 | $(1-k)(c-F)r$ | 0 |
| Film and television enterprises | $aq(p-c)-(aqc-F)r$ | $q\frac{F}{c}(p-c)$ | $q(g-c)-(qc-F)r$ | 0 |
| Negative Completion | $(1-a)q(p-c)$ | $(1-\frac{F}{c})q(p-c)$ | $q(p-g)$ | 0 |
| $1-x$ | $(aqc-F)r$ | 0 | $(1-k)(qc-F)r$ | 0 |

The expected return at negative completion is:

$$E_{x2} = yzq(p-c)(a-\frac{F}{c}) - zr(qc-F)(1-y) + yq\frac{F}{c}(p-c) + zq(1-y)(g-c) - yzr(aqc-F) \tag{2}$$

The average expected return is: $E_x = xE_{x1} + (1-x)E_{x2}$.

At this point, the equation for the replication dynamics of film and television enterprises is:

$$F(x) = x(1-x)[yz(p-c)(a-\frac{F}{c})(1-q) - zrc(1-y)(1-q) + y\frac{F}{c}(p-c)(1-q) + z(1-y)(g-c)(1-q) - yzrac(1-q)] \tag{3}$$

Let $g_1(y) = (1-q)[yz(p-c)(a-\frac{F}{c}) - zrc(1-y) + y\frac{F}{c}(p-c) + z(1-y)(g-c) - yzrac]$.

By constructing the equation for the replication dynamics of film and television enterprises and analyzing its properties, the stable strategy of film and television enterprises can be obtained. From the stability theory [35], if and only if $F(x) = 0, F'(x) < 0$, the strategy of film and television enterprises tends toward a steady state, as shown in Proposition 1.

Proposition 1

(1) When $g_1(y) = 0$, any $x$ is evolutionarily stable.

(2) When $g_1(y) > 0$, the evolutionarily stable strategy for film and television enterprises is $x = 1$, i.e., the film and television enterprises choose positive completion.

(3) When $g_1(y) < 0$, the evolutionary stabilization strategy for film and television enterprises is $x = 0$, i.e., the film and television enterprises choose negative completion.

*Proof.* From the equation for the replication dynamics of film and television enterprises, the first-order derivatives is:

$$F'_x(x) = (1-2x)[yz(p-c)(a-\frac{F}{c})(1-q) - zrc(1-y)(1-q) + y\frac{F}{c}(p-c)(1-q) + z(1-y)(g-c)(1-q) - yzrac(1-q)]. \tag{4}$$

If $g_1(y) = 0$, there is always $F(x) = 0$. Thus, taking any $x$ is a stable strategy. If $g_1(y) > 0, F'_x(x = 1) < 0, F'_x(x = 0) > 0$, then $x = 1$ isa stable strategy; if $g_1(y) < 0, F'_x(x = 1) > 0, F'_x(x = 0) < 0$, then $x = 0$ is a stable strategy. The proof is complete.

According to Proposition 1, when the sales platform and bank strategies satisfy a particular equilibrium condition ($g_1(y) = 0$), the film and television enterprises will not change their existing strategies. When the sales platform and bank strategies satisfy $g_1(y) > 0$, the film and television enterprises will choose the positive completion strategy; conversely, the film and television enterprises will choose the negative completion. $g'_{1a}(y) = (1-q)yz[p - c(1 + r)] > 0$, which suggests that it is an increasing function of $a$, and that an increase in the proportion of the film and television enterprises' revenue-sharing ratio will motivate the film and television enterprises to evolve toward a positive completion strategy.

## 4.2. Strategic stability analysis of sales platforms

The expected returns when the sales platform chooses to cooperate is:

$$E_{y1} = xz(p-c)\left(\frac{F}{c} - a\right)(1-q) + x(1-q)\left(1 - \frac{F}{c}\right)(p-c) + zq(p-c)\left(\frac{F}{c} - a\right) + \left(1 - \frac{F}{c}\right)q(p-c) \tag{5}$$

The expected return when the sales platform chooses not to cooperate is $E_{y2} = xz(1-q)(p-g) + zq(p-g)$.

The average expected return is: $E_y = yE_{y1} + (1-y)E_{y2}$.

At this point, the equation for the replication dynamics of the sales platform is:

$$F(y) = y(1-y)[xz(p-c)(F/c-a)(1-q) + x(1-q)(1-F/c)(p-c) + zq(p-c)(\frac{F}{c} - a) \\ +(1-F/c)q(p-c) - xz(1-q)(p-g) - zq(p-g)], \tag{6}$$

Let $g_2(z) = xz(p-c)(F/c-a)(1-q) + x(1-q)(1-F/c)(p-c) + zq(p-c)(F/c-a)$ $+(1-F/c)q(p-c) - xz(1-q)(p-g) - zq(p-g)$.

By constructing the replication dynamic equation for the platform strategy and analyzing its properties, the stabilization strategy of the platform can be obtained, as shown in Proposition 2.

Proposition 2

(1) When $g_2(z) = 0$, any $y$ is an evolutionary stable state.

(2) When $g_2(z) > 0$, the platform's evolutionary stable strategy is $y = 1$, i.e., the platform chooses to cooperate.

(3) When $g_2(z) < 0$, the platform's evolutionary stable strategy is $y = 0$, i.e., the platform chooses not to cooperate.

*Proof.* From the equation for the replication dynamics of the sales platform, the first-order derivatives are:

$$F'_y(y) = (1-2y)[xz(p-c)(F/c-a)(1-q) + x(1-q)(1-F/c)(p-c) + zq(p-c)(F/c-a) \\ +(1-F/c)q(p-c) - xz(1-q)(p-g) - zq(p-g)]. \tag{7}$$

If $g_2(z) = 0$, there is always $F(y) = 0$, so taking any $y$ is a stable strategy. If $g_2(z) > 0$, $F'_y(y = 1) < 0$, $F'_y(y = 0) > 0$, then $y = 1$ is a stable strategy; if $g_2(z) < 0$, $F'_y(y = 1) > 0$, $F'_y(y = 0) < 0$, then $y = 0$ is a stable strategy. The proof is complete.

According to Proposition 2, when the strategies of film and television enterprises and banks satisfy a particular equilibrium condition ($g_2(z) = 0$), the sales platform does not change its existing strategy. When $g_2(z) > 0$, the sales platform chooses to cooperate; conversely, the sales platform chooses not to cooperate. $g'_{2g}(z) = z(x(1-q) + q) > 0$, indicating an increasing function of $g$. As the negotiating power of film and television enterprises increases, the copyright acquisition cost increases, and the sales platforms evolve towards cooperative strategies. $g'_{2z}(z) = [x(1-q) + q][(p-c)(F/c - a) - (p-g)] < 0$, which indicates a decreasing function of $z$. This also implies that, as the bank's willingness to lend increases, the sales platform evolves towards a non-cooperative strategy. When the bank's willingness to lend is high, it will make the film and television enterprises occupy a higher proportion of the cost input, and then the sales platform tends to choose the non-cooperative strategy.

## 4.3. Strategic stability analysis of banks

The bank's expected return when choosing loans is:

$$E_{z1} = xyrac(1-q) + y(aqc - F)r + x(1-k)rc(1-q) + xyrc(1-k)(1-q) + r(1-y)(1-k)(qc - F). \tag{8}$$

The bank's expected return when it chooses not to lend is: $E_{z2} = 0$.

The average expected return is: $E_z = zE_{z1} + (1-z)E_{z2}$.

At this point, the equation for the replication dynamics of the sales platform is:

$$F(z) = z(1-z)[xyrac(1-q) + y(aqc - F)r + x(1-k)rc(1-q) + xyrc(1-k)(1-q) + r(1-y)(1-k)(qc - F)], \tag{9}$$

Let $g_3(x) = xyrac(1-q) + y(aqc - F)r + x(1-k)rc(1-q) + xyrc(1-k)(1-q) + r(1-y)(1-k)(qc - F)$.

The bank's stabilization strategy can be obtained by constructing a replicated dynamic equation for the bank's strategy and analyzing its properties, as shown in Proposition 3.

Proposition 3

(1) When $g_3(x) = 0$, any $z$ is evolutionarily stable.

(2) When $g_3(x) > 0$, the bank's evolutionarily stable strategy is $z = 1$, i.e., the bank chooses to lend.

(3) When $g_3(x) < 0$, the bank's evolutionarily stable strategy is $z = 0$, i.e., the bank chooses not to lend.

*Proof.* From the equation for the replication dynamics of the sales platform, the first-order derivatives are:

$$F'_z(z) = (1-2z)[xyrac(1-q) + y(aqc - F)r + x(1-k)rc(1-q) + xyrc(1-k)(1-q) + r(1-y)(1-k)(qc - F)]. \tag{10}$$

If $g_3(x) = 0$, there is always $F(z) = 0$. Therefore, taking any $z$ is a stable strategy. $g_3(x)$ is constantly greater than 0. Only when $z = 1$, $F'_z(x) < 0$, indicating that banks always tend to choose the lending strategy. This is due to the fact that in the model constructed in this paper, it is assumed that bank loans can achieve positive returns, and in the case of the sales platform choosing not to cooperate, the bank only needs to pay a certain proportion of the management cost, while the profit is 0 in the case of the bank not lending, so the bank always tends to choose loans. The proof is complete.

According to Proposition 3, when the strategy of the film and television enterprises and the sales platform satisfy a particular condition ($g_3(x) = 0$), the bank will not change its existing strategy. $g'_{3x}(x) = [yrac(1-q) + (1-k)rc(1-q) + yrc(1-k)(1-q)] > 0$, indicating an increasing function of $x$. This also implies that banks tend to choose to provide loans when film and television enterprises positively complete the production of their films. $g'_{3F}(x) = -yr - r(1-y)(1-k) < 0$, which indicates a decreasing function of $F$, suggesting that, as the own funds of

film and television enterprises diminish, banks are more likely to choose to provide loans. Film and television enterprises choose positive completion strategies and borrow more money from the bank, the bank profits improve; banks choose to loan, and film and television enterprises can also benefit through the cost-sharing ratio of the enhancement of the proportion of income, with access to a greater proportion, and the two are in a mutually reinforcing state. In real life, should also be as far as possible to achieve the bank chooses to loan, film and television enterprises choose to positive completion of the state. For banks, the funds of film and television enterprises do not present a key factor affecting their decision-making, but rather less funds can inspire the bank's willingness to lend.

### 4.4. Analysis of evolutionary stabilization strategies for game tripartite

Based on the replicated dynamic equations of the three parties of the game in the previous section, the Jacobi matrix of the three-party evolutionary game system can be further obtained, i.e.,:

$$J = \begin{bmatrix} \frac{\partial F(x)}{\partial x} & \frac{\partial F(x)}{\partial y} & \frac{\partial F(x)}{\partial z} \\ \frac{\partial F(y)}{\partial x} & \frac{\partial F(y)}{\partial y} & \frac{\partial F(y)}{\partial z} \\ \frac{\partial F(z)}{\partial x} & \frac{\partial F(z)}{\partial y} & \frac{\partial F(z)}{\partial z} \end{bmatrix}. \tag{11}$$

Let $F(x) = 0, F(y) = 0, F(z) = 0$, and we can get 8 local equilibrium points: $E_1(0, 0, 0), E_2(0, 0, 1), E_3(0, 1, 0), E_4(0, 1, 1), E_5(1, 0, 0), E_6(1, 0, 1), E_7(1, 1, 0), E_8(1, 1, 1)$, where the elements in parentheses correspond to the values of the strategies $(x, y, z)$ that should be adopted by the film and television enterprises, sales platforms, and banks in this equilibrium point, respectively. According to Lyapunov's law [36], when the eigenvalues of the Jacobian matrix are all negative, the local equilibrium point is the evolutionary stable strategy (ESS). The eigenvalues of the Jacobian matrix corresponding to each equilibrium point are calculated as shown in Table 3.

Through Table 3, $E_2(0, 0, 1), E_6(1, 0, 1), E_8(1, 1, 1)$ may be a stabilization point in the game model.

Proposition 4

(1) When $g > ap + c - ac$, there is a stabilization point $E_8(1, 1, 1)$. The corresponding scenarios are film and television enterprises choosing to positively complete the production of their films, sales platforms choosing to cooperate with film and television enterprises, and banks choosing to provide loans.

**Table 3. Equilibrium point stability analysis.**

| Equilibrium point | Eigenvalue 1 | Eigenvalue 2 | Eigenvalue 3 | Stability |
|---|---|---|---|---|
| $E_1(0, 0, 0)$ | 0 | $(1 - F/c)q(p - c)$ | $r(1 - k)(qc - F)$ | Instability |
| $E_2(0, 0, 1)$ | $(g - c)(1 - q) - rc(1 - q)$ | $q(p - c)(1 - a) - q(p - g)$ | $-r(1 - k)(qc - F)$ | – |
| $E_3(0, 1, 0)$ | $F/c(p - c)(1 - q)$ | $-(1 - F/c)q(p - c)$ | $(aqc - F)r$ | Saddle point |
| $E_4(0, 1, 1)$ | $(p - c)a(1 - q) - rac(1 - q)$ | $q(p - g) - q(p - c)(1 - a)$ | $-(aqc - F)r$ | Saddle point |
| $E_5(1, 0, 0)$ | 0 | $(1 - F/c)(p - c)$ | $(1 - k)rc(1 - q) + r(1 - k)(qc - F)$ | Instability |
| $E_6(1, 0, 1)$ | $rc(1 - q) - (g - c)(1 - q)$ | $(1 - a)(p - c) - (p - g)$ | $-(1 - k)rc(1 - q) - r(1 - k)(qc - F)$ | – |
| $E_7(1, 1, 0)$ | $-F/c(p - c)(1 - q)$ | $-(1 - F/c)(p - c)$ | $rac(1 - q) + (aqc - F)r + 2(1 - k)rc(1 - q)$ | Saddle point |
| $E_8(1, 1, 1)$ | $rac(1 - q) - a(p - c)(1 - q)$ | $(p - g) - (p - c)(1 - a)$ | $-rac(1 - q) - (aqc - F)r - 2rc(1 - k)(1 - q)$ | – |

Note: '-' indicates uncertainty of stability at this point

(2) When $g < ap + c - ac$ and $g > c(1 + r)$, there exists a stability point $E_6(1, 0, 1)$. In the corresponding scenario, film and television enterprises chooses to positively complete the production of the film, the sales platform chooses not to cooperate with them, and the bank chooses to provide loans.

(3) When $g < ap + c - ac$ and $g < c(1 + r)$, there is a stabilization point $E_2(0, 0, 1)$. In the corresponding scenario, film and television enterprises choose to negatively complete the production of the film, the sales platform chooses not to cooperate, and the bank chooses to provide loans.

*Proof*. When $g > ap + c - ac$, $(1 - a)(p - c) > p - g$, sales platforms achieve higher profits when they choose to work with film and television enterprises. Based on the previous assumption that $p > c(1 + r)$, $rac(1 - q) - a(p - c)(1 - q) < 0$, at this point, all three eigenvalues are negative and are stable points. When $g < ap + c - ac$, $(1 - a)(p - c) < p - g$, sales platforms achieve higher profits when they choose not to cooperate. When $g > c(1 + r)$, $rc(1 - q) - (g - c)(1 - q) < 0$, all three eigenvalues are negative and are stable points. When $g < c(1 + r)$, $(g - c)(1 - q) - rc(1 - q) < 0$, all three eigenvalues are negative and are stable points. The proof is complete.

Through the above propositions, it can be seen that different costs of copyright acquisition, film production costs, and revenue-sharing ratios will result in different stable points in the evolutionary game. A linear graph is used to illustrate the influence of these parameters on each other and their impact on the overall stable points. This leads to Fig 1. By observing Fig 1 below, it can be determined that:

(1) The decrease of the equity ratio of film and television enterprises (*a* decrease) is positively correlated with the possibility of the sales platform choosing cooperation strategy. The lower the equity ratio of film and television enterprises, the higher the possibility of cooperation of the platform; conversely, the lower the possibility of cooperation.

(2) When film and television enterprises show strong negotiation ability, making the copyright acquisition cost relatively high, the sales platform will consider establishing cooperative relationship with them. With the gradual increase of film production costs, film and television enterprises must further improve their negotiation ability to prompt sales platforms to make cooperation decisions.

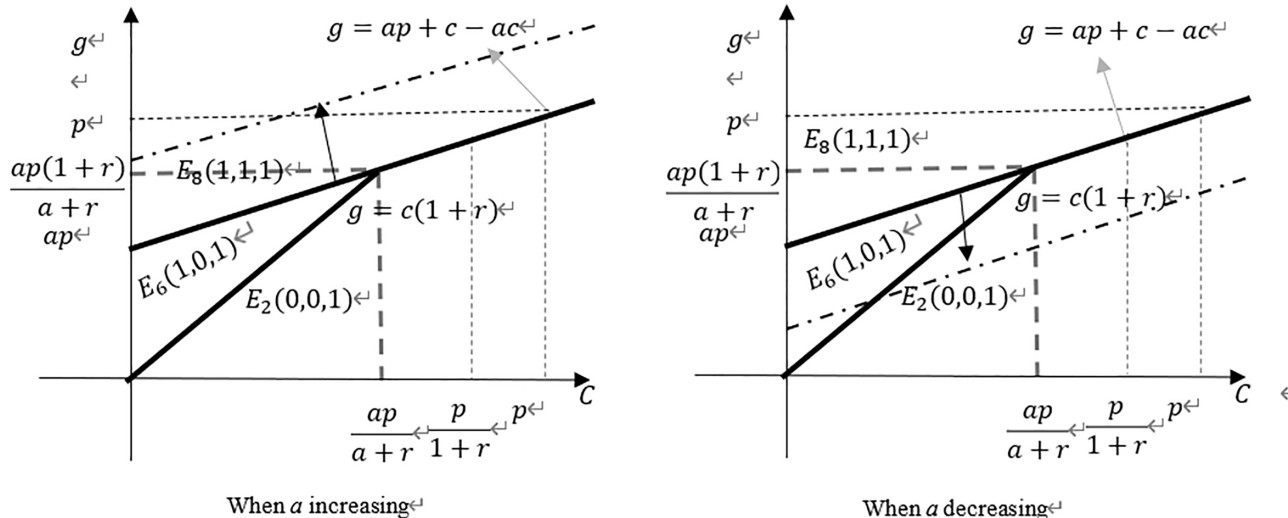

**Fig 1. Stability point strategy analysis diagram.**

(3) When the sales platform chooses to cooperate, film and television enterprises will choose positive completion. When the sales platform does not cooperate, film and television enterprises are limited in funds. If the production cost of the film is low, the film and television enterprises will choose to positively complete the production of the film with the help of bank financing. If the production cost of the film is high, the film and television enterprises have significant financial constraints, and they will choose the negative completion strategy.

## 5. Simulation analysis

Through the above analysis, the final stabilization strategies of the three-party evolutionary game model of film and television enterprises, sales platforms, and banks under different conditions are given, but the specific evolutionary game process cannot be demonstrated. To analyze the evolution process of the three-party game more intuitively and concisely under different conditions, the choice of strategies of the participants in the initial state and the influence of different parameter changes on the evolutionary stable strategy. This part uses MATLAB to numerically simulate the three-party evolutionary game model of film and television enterprises, sales platforms, and banks. Based on the data of Huayi Brothers Research Institute [37], for a Hollywood film with proceed of $100 million, the average production cost is $30,482,379 and the average miscellaneous cost is $14,828,500, which means the operations cost as a percentage of revenue are about 40%; the average copyright acquisition cost is $51,642,345, which means the copyright cost as a percentage of revenue are about 50%. Based on the data of [38] a classic Hollywood film's proceed is about $400 million. Thus, we set $p = 400$, and the copyright acquisition cost is $g = 200$, which is 50% of the proceed, and the Costs incurred by film and television enterprises for making the film is $c = 150$, which is 37.5% of the proceed. The bank loan interest rate is assumed to be $r = 0.05$, which is in line with reality. Other parameters are $a = 0.3, q = 0.3, F = 10$ and $k = 0.4$. The initial state of the tripartite strategy selection $(x, y, z)$ is set to $(0.5, 0.5, 0.5)$. To analyze the tripartite evolutionary stabilization strategies under different conditions, the parameter settings are adjusted to satisfy the conditions for subsequent numerical analysis.

### 5.1. Analysis of evolutionary processes under different conditions

On the basis of the basic parameter settings, to satisfy proposition 4(1), adjust $g = 250, g = 350$, the stability point of the evolutionary game is $E_8(1, 1, 1)$, corresponding to the scene for the film and television enterprises choose to positive completion of the film production, the sales platform to choose to cooperate, and the bank to choose to loan. By observing Fig 2, when the parameters under the condition of Proposition 4(1) are satisfied, film and television enterprises tend to choose positive completion of film production, sales platforms tend to cooperate, and banks tend to lend. As $g$ increases, the faster the sales platform tends to choose cooperation and the stronger the willingness to cooperate.

Based on the basic parameter settings, propositions 4(2) and 4(3) are satisfied, with adjustment of $g = 100$ and $g = 175$. At this time, the stability points of the evolutionary game are $E_2(0, 0, 1)$ and $E_6(1, 0, 1)$, and the corresponding scenarios are film and television enterprises choosing the negative completion of the film production, sales platforms choosing non-cooperation, the bank choosing loans, and film and television enterprises choosing the positive completion of the film production, sales platform choosing non-cooperation, the bank chooses loans. By observing Fig 3, when film and television enterprises have lower negotiation ability, the copyright acquisition cost will be smaller, and the sales platform will eventually evolve to non-cooperation strategy over time. For banks, the level of copyright acquisition cost does not affect their choice of strategy.

### 5.2. System evolution paths for optimal policy combinations under different initial states

To satisfy proposition 4(1), the parameters are set as $a = 0.3, p = 400, c = 150, F = 10, r = 0.05, q = 0.3, g = 250, k = 0.4$. At this time, the stability point of the evolutionary game is $E_8(1, 1, 1)$, the corresponding scenario is that film and television

                                   

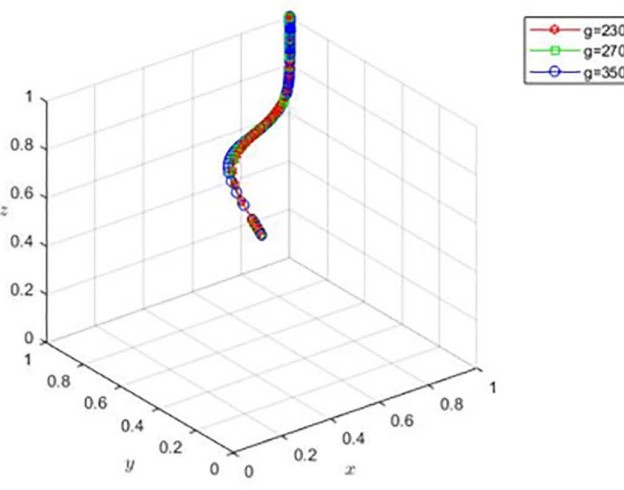
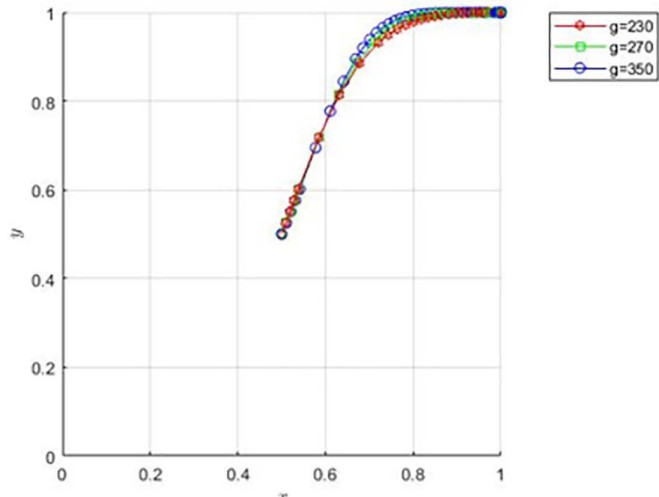

**Fig 2. Evolution when procurement costs are large.**

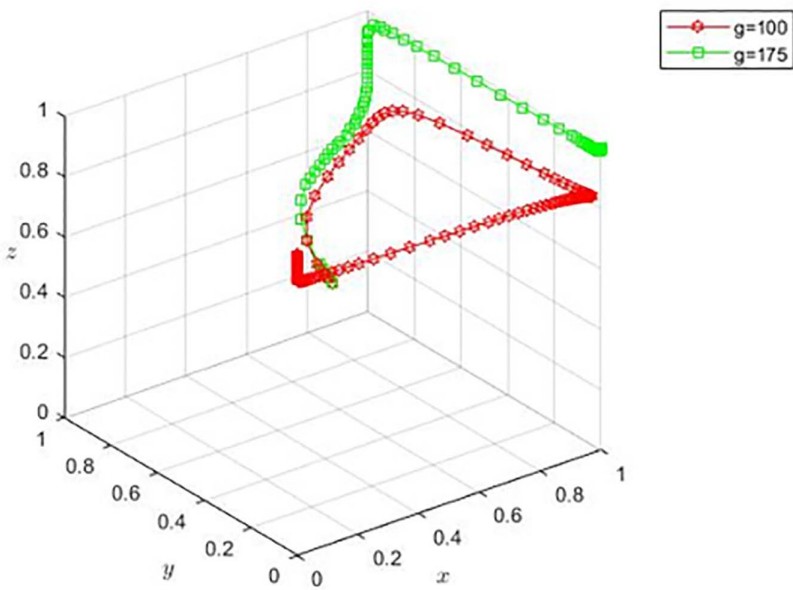

**Fig 3. Evolution when procurement costs are small.**

enterprises choose to positively complete the production of the film, the sales platform chooses to cooperate, and the bank chooses to take out loans. This is the ideal state that film and television enterprises want to achieve before the shooting of the movie, which is also the focus of the article's research. This section focuses on considering the system evolution path of Proposition 4(1) under different initial states.

Based on the parameter settings of the basic model, the conditions of Proposition 4(1) are satisfied to analyze the situation in which film and television enterprises positively complete film production, sales platforms choose cooperation, and banks choose loans with low initial willingness, at which time the state of $(x, y, z)$ strategy selection is $(0.3, 0.3, 0.3)$

; It also analyzes the situation in which film and television enterprises positively complete film production, sales platforms choose cooperation, and banks choose loans with high initial willingness. The strategy selection state is $(0.7, 0.7, 0.7)$. By observing Fig 4, it can be found that regardless of the initial willingness of film and television enterprises to choose positive completion, sales platforms to choose cooperation, and banks to choose loans, all of them will evolve towards the stable point $E_8(1, 1, 1)$ with time under the fulfillment of certain conditions. This also suggests that, if film and television enterprises are willing to put in more effort to improve the profits of the sales platforms and the banks, they can eventually reach the ideal state.

### 5.3. Effect of different parameters on the optimal strategy portfolio

In this paper, the stabilization point $E_8(1, 1, 1)$ is the optimal strategy, and the corresponding scenario is that film and television enterprises choose positive completion of film production, sales platforms choose cooperation, and banks choose loans. Next, this paper analyzes the degree of influence of different parameters on the optimal strategy through the changes of key parameters.

Based on the parameter design of the basic model, $g = 100, g = 300$ is applied to analyze the evolutionary strategy choices of different costs of copyright acquisition for film and television enterprises, sales platforms, and banks. Fig 5 shows that, for the sales platform, the increase in copyright acquisition cost caused by the enhancement of the negotiation ability of film and television enterprises will drive the sales platform to choose the cooperation strategy. In contrast, the sales platform evolves into a non-cooperative strategy. For film and television enterprises, when the sales platform tends to choose cooperation and other conditions remain unchanged, film and television enterprises are more inclined to choose the positive completion of film production.

Based on the parameter design of the basic model, $g = 300, F = 10, F = 80$ and $F = 200$ are taken to analyze the evolution strategy choices of the different funds of film and television enterprises, sales platforms, and banks. The blue and red lines in Fig 6 indicate that, when the funds of film and television enterprises are insufficient, the higher procurement cost prompts the sales platform to choose to cooperate, the change of their funds will not affect the strategic choices

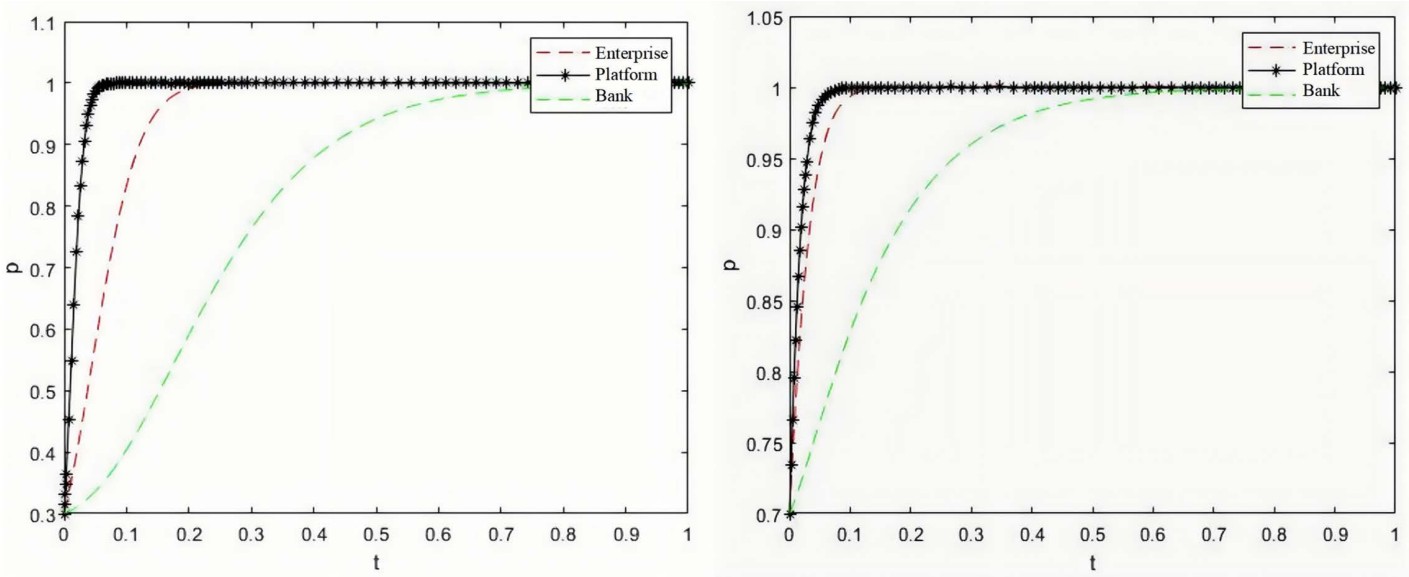

**Fig 4. Tripartite evolution process under different initial states.**

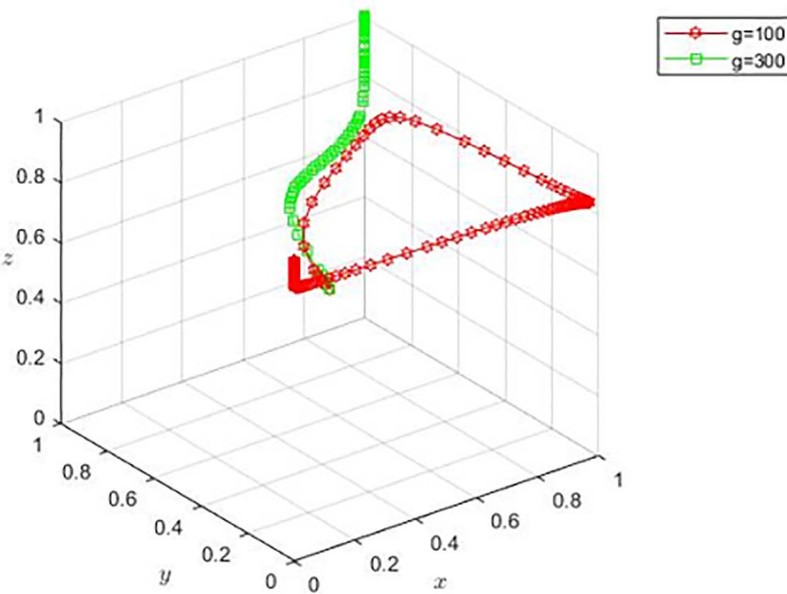

**Fig 5. Effect of different purchasing costs on the optimal strategy.**

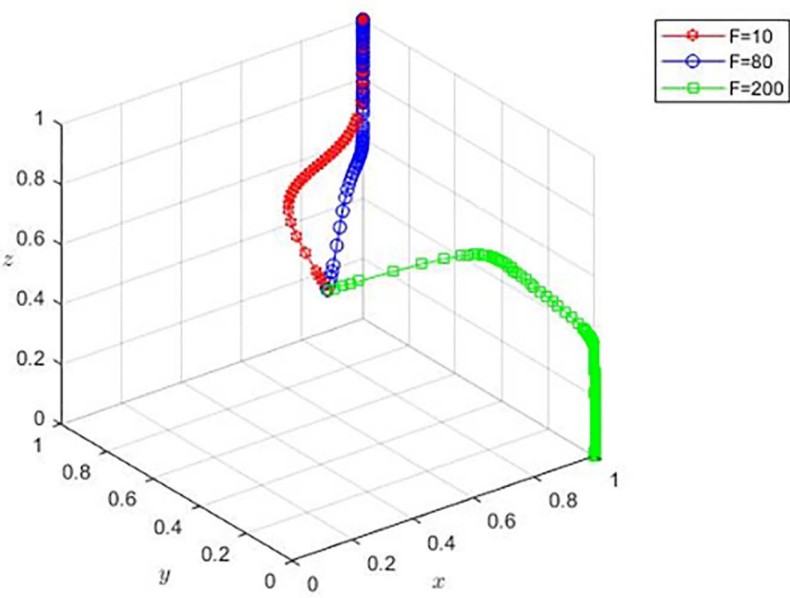

**Fig 6. Effect of different own funds on the optimal strategy.**

of film and television enterprises, and the film and television enterprises will always choose to positively complete the production of the film. The lower funds of film and television enterprises will make banks choose to provide loans. The green line in the figure shows that the sales platform will choose the non-cooperation strategy when the negotiation ability of film and television enterprises is high, and the funds of film and television enterprises are sufficiently large. This is

because, when film and television enterprises have sufficient funds, for sales platforms, the proportion of equity gained when choosing to cooperate is lower, thus they will gradually transform to the non-cooperation strategy. Similarly, when film and television enterprises have lower owned funds, banks will always choose to provide loan strategy, and when the film and television enterprises have more funds, the banks' likelihood to supply loan profit is relatively reduced. In the sale of platforms to choose not to cooperate, the bank also needs to manage the amount of the loan, so it will gradually to the direction of the evolution of the non-loan.

## 6. Conclusion

This paper analyzes the stability of the strategic choices of each party and the influence of some factors on the evolution strategy by constructing the three-party evolution game model of film and television enterprises, sales platforms, and banks. It also carries out numerical simulation to draw the following conclusions:

(1) With the increase of equity ratio and the bank's willingness to lend, film and television enterprises are more inclined toward positive completion of film production. This is consistent with the actual situation, when the film and television enterprises face insufficient funds. If they can obtain bank loan support, coupled with a higher proportion of income distribution, the enterprises will be more motivated to devote themselves to the filming of the film to ensure the successful completion of the film.

(2) The copyright acquisition cost of movie and television rights is the key for sales platforms to make strategic choices. If the negotiation power of film and television enterprises is weak, the sales platform will have a greater advantage in the negotiation process and may acquire copyrights at a lower cost. Thus they tend to choose not to cooperate with the film and television enterprises. In addition, the willingness of bank loans and the proportion of equity of film and television enterprises have an inverse relationship with the willingness of sales platforms to cooperate, which is because both will lead to the sales platforms obtaining a smaller share of the proceeds in the process of cooperation.

(3) There is a positive relationship between the positive completion of film and television enterprises and bank loan willingness. In addition, the article also revealed an interesting phenomenon, that is, for film and television enterprises, greater amounts of owned funds of film and television enterprises are not better; only when the amount of own funds exceeds a certain limit does it have a positive impact. A relatively small amount of capital tends to attract banks' willingness to lend.

The findings of this paper bring certain management insights to the subsequent tripartite cooperation among film and television enterprises, sales platforms, and banks.

In the processing of film and television, the acquisition of capital and market resources is particularly critical. Through close cooperation with banks and sales platforms, film and television enterprises can better solve their core problems and promote the smooth progress of filming. In terms of cooperation with banks, film and television enterprises should be not limit to the scale of their existing capital, but also enhance their attractiveness and influence through various ways. By producing high-quality film and television works with market potential, they can win the trust and favor of banks. As a distribution and promotion channel for film and television works, a sales platform is crucial for the development of film and television enterprises. In cooperation with sales platforms, film and television enterprises can obtain financial and resource support from sales platforms by transferring part of their equity to jointly promote the distribution and promotion of their works. However, sales platforms should also realize that, although buying copyrights at a low price may bring certain benefits in the short term, in the long term, a stable partnership and a mutually beneficial cooperation model are the keys to promoting the continued prosperity of the film and television industry. Through the cooperation mode of risk-sharing and benefit-sharing such as co-production, the risk borne by a single party can be effectively reduced, and the quality and social influence of the whole project can be enhanced. Therefore, sales platforms should focus on establishing long-term

and stable cooperative relationships with film and television enterprises and jointly promote the prosperity of the film and television industry through a mutually beneficial and win-win cooperation model.

Based on the strategic choices of film and television enterprises, sales platforms, and banks, this paper constructs a three-party evolutionary game model. It explores the strategic choices of film and television enterprises, sales platforms, and banks when there are financial constraints for film and television enterprises in the process of copyright acquisition. The owned funds of film and television enterprises are an important factor that affects the behavior of the three strategic choices. This paper assumes that the bank can always obtain principal and interest. Future research could also consider the bank in a certain situation in which it can't obtain principal and interest, and strategic choices among the three are made.

## Supporting information

**S1 File. Code file for the experimental section.**
(DOCX)

## Author contributions

**Conceptualization:** Wenli Wang.

**Data curation:** Hui Li.

**Formal analysis:** Ye Zhen.

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
