## [Decision Letter · Decision Letter 0]

2 Sep 2024

Dear Dr. 李,

Thank you for submitting your manuscript to PLOS ONE. After careful consideration, we feel that it has merit but does not fully meet PLOS ONE’s publication criteria as it currently stands. Therefore, we invite you to submit a revised version of the manuscript that addresses the points raised during the review process.

I am returning your manuscript with four reviews. The reviewers arrived at different conclusions about the paper, as you will see. After reading the reviews and looking at the manuscript, I am afraid that I have to concur with the more critical review.

I am sorry I cannot be more positive at the moment, but as I have noted, all is not lost. It needs a minor revision, and if you wish to resubmit, I believe you should work on the manuscript.<o:p></o:p>

Note that it will have to go through the second round of review. Please pay attention to the reviewer suggestions and give them due consideration.

We look forward to receiving your revised manuscript.

Kind regards,

Godwin Banafo Akrong, Ph.D.

Academic Editor

PLOS ONE

“This work was supported by the< The Youth Fund for Humanities and Social Sciences Research of the Ministry of Education, "Research on Blockchain Based Digital Copyright Pledge Financing Model and Risk Management Mechanism"> under Grant [number 21YJC630172]; < Project of National Natural Science Foundation of China: Research on Risk Transmission Mechanism and Prevention and Control Strategy of Supply Chain Finance under Major Emergencies > under Grant [number 72171162]; and < Youth Fund for Humanities and Social Sciences Research of Ministry of Education "Research on Innovative Financing Mechanism of Order Agricultural Supply Chain Supported by Commercial Insurance" > under Grant [number 20YJC630148]”

Reviewers' comments:

Reviewer's Responses to Questions

**Comments to the Author**

1. Is the manuscript technically sound, and do the data support the conclusions?

Reviewer #1: Yes

Reviewer #2: Yes

Reviewer #3: Yes

Reviewer #4: Yes

2. Has the statistical analysis been performed appropriately and rigorously?

Reviewer #1: Yes

Reviewer #2: Yes

Reviewer #3: N/A

Reviewer #4: Yes

3. Have the authors made all data underlying the findings in their manuscript fully available?

Reviewer #1: Yes

Reviewer #2: Yes

Reviewer #3: No

Reviewer #4: No

4. Is the manuscript presented in an intelligible fashion and written in standard English?

Reviewer #1: Yes

Reviewer #2: Yes

Reviewer #3: Yes

Reviewer #4: Yes

Reviewer #1: The write up is excellent. The stage chosen is for film but can be easily applied to banking financing of other sectors. Paper is well written I have nothing more to add other than it would be good if the model is tested on actual data.

Reviewer #2: The paper presents a tripartite evolutionary game model involving film and television enterprises, sales platforms, and banks and discusses the strategic choices of operation and the investment and financing behaviors of the three parties.It has a good structure and follows a scientific approach. The results are well interpreted, however, they lack discussion with other findings.

Reviewer #3: Conclusively, my suggestion for this article is to accept it after a minor revision. This article has no methodological errors. However, I suggest the author provides the following revisions:

1. The literature review of this manuscript did not show the advantage of applying an evolutionary model to the studied problem. I suggest the author present this advantage for illustrating why an evolutionary model was employed in this article.

2. I further recommend the author reviews the theoretical background of an evolutionary model. Just applying an evolutionary model is insufficient.

3. How to plot Figure 1 on the 10th page of this manuscript was not sufficiently explained. Please add the explanation.

4. Are baseline models available for this article? I suggest the author may provide some baseline models.

Reviewer #4: Referee report on “EVOLUTIONARY GAME ANALYSIS OF SUPPLY CHAIN FINANCING FOR FILM AND TELEVISION ENTERPRISES CONSIDERING CO-PRODUCTION”

Summary

Currently, many film and television enterprises struggle with insufficient funds during the visualization process. To address this issue, companies can opt to enter into coproduction agreements with sales platforms, implement supply chain financing, obtain financing through banks, or collaborate with all three parties simultaneously. The paper presents a tripartite evolutionary game model involving film and television enterprises, sales platforms, and banks and discusses the strategic choices of operation and the investment and financing behaviors of the three parties. Additionally, it analyzes film and television enterprises' capital, film production costs, copyright acquisition costs, and other factors and examines the impact of these factors on their strategic choices. The research provides theoretical support for further alleviating the financing difficulties of film and television enterprises and at the same time provides a theoretical basis for the cooperation among film and television enterprises, sales platforms, and banks. This topic is interesting and the work is carefully designed. However, there are still concerns as follows.

Major concerns:

1. In the introduction section, it seems more discussion is expected to explain the contribution of this work and the novel findings.

2. In section 2, there shall be the literature review which contains discussions about film and television enterprises.

3. In section 3, there is no discussion of the existing literature on model assumptions and the reasons why this article focuses on this design.

Minor concerns:

1. The formatting of the conclusion is a bit strange, which requires revision.

**Do you want your identity to be public for this peer review?** For information about this choice, including consent withdrawal, please see our Privacy Policy

Reviewer #1: No

Reviewer #2: No

Reviewer #3: **Yes: ** Kuang Yih Hsu

Reviewer #4: No

---

## [Author Response · Author response to Decision Letter 1]

27 Nov 2024

Response to Reviewers

We acknowledge the reviewers’ comments which are valuable in improving the quality of our manuscript. We have revised our manuscript addressing these comments or suggestions. The changes in the revised version are colored blue.

Reviewer 1:

The write up is excellent. The stage chosen is for film but can be easily applied to banking financing of other sectors. Paper is well written I have nothing more to add other than it would be good if the model is tested on actual data.

Response: Thank you very much for the insightful comments regarding the model validation. We fully agree that using real data to test the model can further enhance the empirical foundation of the research. Thus, in the section of “4. Simulation analysis”, we have added the actual data to explain the rationality of parameter settings as follows:

“Based on the data of Huayi Brothers Research Institute (2017), for a Hollywood film with proceed of $100 million, the average production cost is $30,482,379 and the average miscellaneous cost is $14,828,500, which means the operations cost as a percentage of revenue are about 40%; the average copyright acquisition cost is $51,642,345, which means the copyright cost as a percentage of revenue are about 50%. Based on the data of Weisaidi (2021), a classic Hollywood film’s proceed is about $400 million. Thus, we set and the copyright acquisition cost is , which is 50% of the proceed, and the Costs incurred by film and television enterprises for making the film is , which is 37.5% of the proceed. The bank loan interest rate is assumed to be , which is in line with reality.”

Reviewer 2:

The paper presents a tripartite evolutionary game model involving film and television enterprises, sales platforms, and banks and discusses the strategic choices of operation and the investment and financing behaviors of the three parties.It has a good structure and follows a scientific approach. The results are well interpreted, however, they lack discussion with other findings.

Response: Thank you very much for the valuable suggestion. We fully agree with your perspective that the discussion of our current results in relation to other relevant findings could be further strengthened. In the revised manuscript, the introduction section has been rewritten to re-examine the contributions and novel discoveries of the article. The literature review section has been expanded to include an overview of other studies that are relevant to or comparable with our research findings, as well as a discussion on the advantages of the research methodology employed in this paper.

Reviewer 3: Conclusively, my suggestion for this article is to accept it after a minor revision. This article has no methodological errors. However, I suggest the author provides the following revisions:

1. The literature review of this manuscript did not show the advantage of applying an evolutionary model to the studied problem. I suggest the author present this advantage for illustrating why an evolutionary model was employed in this article.

Response: Thank you very much for the valuable suggestion. In the revised version of this paper, the literature review section has been updated to articulate the distinctive advantages of introducing evolutionary game theory models into the research of film and television industry financing. The evolutionary model not only transcends the static constraints of traditional financing analysis frameworks but also delineates the evolutionary mechanisms of financing strategies employed by film and television enterprises. This approach offers a novel perspective and strategic recommendations for addressing the prevalent issues of difficult and expensive financing in the film and television industry.

2. I further recommend the author reviews the theoretical background of an evolutionary model. Just applying an evolutionary model is insufficient.

Response: Thank you very much for the valuable suggestion. In the introduction section, the valuable feedback and insights provided by the reviewer experts have been incorporated, leading to a detailed elaboration of the theoretical background of evolutionary game theory.

3. How to plot Figure 1 on the 10th page of this manuscript was not sufficiently explained. Please add the explanation.

Response: Thank you very much for pointing out the insufficient explanation of Figure 1 in the paper. In the revised manuscript, I have added a textual description of Figure 1, elaborating on its construction and its relation to the research conclusions, thereby enhancing its clarity and readability.

4. Are baseline models available for this article? I suggest the author may provide some baseline models.

Response: Thank you very much for the suggestion. This article studies the financing issues of film and television companies through an evolutionary game model, a method typically aimed at dynamic environments and strategy evolution processes. In contrast, benchmark models are generally used in relatively stable environments, making direct comparisons with static benchmark models challenging. Therefore, the article does not design a benchmark model.

Reviewer 4:

Referee report on “EVOLUTIONARY GAME ANALYSIS OF SUPPLY CHAIN FINANCING FOR FILM AND TELEVISION ENTERPRISES CONSIDERING CO-PRODUCTION”

Summary

Currently, many film and television enterprises struggle with insufficient funds during the visualization process. To address this issue, companies can opt to enter into coproduction agreements with sales platforms, implement supply chain financing, obtain financing through banks, or collaborate with all three parties simultaneously. The paper presents a tripartite evolutionary game model involving film and television enterprises, sales platforms, and banks and discusses the strategic choices of operation and the investment and financing behaviors of the three parties. Additionally, it analyzes film and television enterprises' capital, film production costs, copyright acquisition costs, and other factors and examines the impact of these factors on their strategic choices. The research provides theoretical support for further alleviating the financing difficulties of film and television enterprises and at the same time provides a theoretical basis for the cooperation among film and television enterprises, sales platforms, and banks. This topic is interesting and the work is carefully designed. However, there are still concerns as follows.

Major concerns:

1. In the introduction section, it seems more discussion is expected to explain the contribution of this work and the novel findings.

Response: Thank you very much for the valuable suggestion. In the revised manuscript, the introduction section has undergone a major revision. To better lead into the contributions and novel discoveries of this paper, the introduction now provides a thorough analysis of the theoretical background of evolutionary game theory. It explicitly outlines the unique contributions of this paper compared to existing studies, including the issues addressed, as well as the new perspectives and methodologies offered.

2. In section 2, there shall be the literature review which contains discussions about film and television enterprises.

Response: Thank you very much for the valuable suggestion. In the revised version, the second section has been adjusted to serve as a comprehensive literature review, which now includes additional relevant literature pertaining to film and television enterprises.

3. In section 3, there is no discussion of the existing literature on model assumptions and the reasons why this article focuses on this design.

Response: Thank you very much for the valuable suggestion. In Section “2.1 Model assumptions” of the revised version, we have added explanations to illustrate the rationality of model assumptions and have cited relevant references to support it.

Minor concerns:

1. The formatting of the conclusion is a bit strange, which requires revision.

Response: I'm very sorry for the inconvenience. I have rearranged and revised the conclusion part according to the requirements of the journal format.

---

## [Decision Letter · Decision Letter 1]

3 Jan 2025

Dear Dr. Zhen,

Thank you for submitting your manuscript to PLOS ONE. After careful consideration, we feel that it has merit but does not fully meet PLOS ONE’s publication criteria as it currently stands. Therefore, we invite you to submit a revised version of the manuscript that addresses the points raised during the review process.

Clarify the contribution of the study (Reviewer #3).Address concerns raised about Reviewer #3 in the Literature Review Section.

In summary, I encourage you to address all the comments raised by Reviewer #3 and make the necessary revisions, particularly in improving the literature review and contributions of the study.

We look forward to receiving your revised manuscript.

Kind regards,

Godwin Banafo Akrong, Ph.D.

Academic Editor

PLOS ONE

Journal Requirements:

Reviewers' comments:

Reviewer's Responses to Questions

**Comments to the Author**

Reviewer #1: All comments have been addressed

Reviewer #2: All comments have been addressed

Reviewer #3: All comments have been addressed

2. Is the manuscript technically sound, and do the data support the conclusions?

Reviewer #1: Yes

Reviewer #2: Yes

Reviewer #3: Yes

3. Has the statistical analysis been performed appropriately and rigorously?

Reviewer #1: Yes

Reviewer #2: Yes

Reviewer #3: N/A

4. Have the authors made all data underlying the findings in their manuscript fully available?

Reviewer #1: Yes

Reviewer #2: Yes

Reviewer #3: No

5. Is the manuscript presented in an intelligible fashion and written in standard English?

Reviewer #1: Yes

Reviewer #2: Yes

Reviewer #3: Yes

Reviewer #1: Now the paper is excellent. Nifty mathematics but in real life things work when there is common interest between all parties. Don’t need mathematics to tell us that.

Reviewer #2: It is evident that the authors have meticulously revised and refined the reviewers' comments, thereby enhancing the quality of the revised manuscript. In my opinion, this renders it highly suitable for immediate publication.

Reviewer #3: Conclusively, my suggestion for this article is to accept it after a minor revision. I suggest the authors provide the following changes:

1. In the literature review of this manuscript, I have read many published papers, which considered the main problem considered in this article.

Nevertheless, these previous papers seem not to be devoted to the game theory. Why did the authors choose the game theory to implement the

current study?

2. The contribution of this article is unclear. It looks lile that the authors just wanted to develop a new application of the game theory. The authors

can define the contribution in the end of the Introduction section.

After complementing the above two questions, the review can end.

**Do you want your identity to be public for this peer review?** For information about this choice, including consent withdrawal, please see our Privacy Policy

Reviewer #1: No

Reviewer #2: No

Reviewer #3: **Yes: ** Guang Yih Sheu

---

## [Author Response · Author response to Decision Letter 2]

13 Jun 2025

We acknowledge the reviewers’ comments which are valuable in improving the quality of our manuscript. We have revised our manuscript addressing these comments or suggestions. The changes in the revised version are colored blue.

Reviewer 1:

Now the paper is excellent. Nifty mathematics but in real life things work when there is common interest between all parties. Don’t need mathematics to tell us that.

Response: Thank you very much for your recognition of our paper. Indeed, in real life, things work when there is common interest between all parties, which is consistent with our game equilibrium. However, the overall interests of the film and television enterprise, the sales platform and the bank under the game equilibrium are not optimal. How to design the financing strategies to achieve a win-win situation for all three parties requires mathematical analysis methods.

Reviewer 2:

It is evident that the authors have meticulously revised and refined the reviewers’ comments, thereby enhancing the quality of the revised manuscript. In my opinion, this renders it highly suitable for immediate publication.

Response: Thank you very much for your valuable suggestions mentioned earlier. Thank you very much for the acceptance of our paper.

Reviewer 3:

Conclusively, my suggestion for this article is to accept it after a minor revision. I suggest the authors provide the following changes:

1. In the literature review of this manuscript, I have read many published papers, which considered the main problem considered in this article.

Nevertheless, these previous papers seem not to be devoted to the game theory. Why did the authors choose the game theory to implement the current study?

Response: Thank you very much for the valuable suggestion. In the revised version of this paper, we have added an overall review of the existing literature and an explanation of why game theory was applied in our study. The specific description is as follows:

“However, most of the existing literature uses qualitative methods such as policy and case studies to analyze the factors influencing the financing of film and television enterprises and the countermeasures to alleviate the problem of financing difficulties for small and medium-sized enterprises; there are also studies using empirical research methods to analyze the factors influencing the success of crowdfunding for film and television enterprises and the paths for film and television enterprises to implement crowdfunding financing. The existing research methods are limited to the decision-making of individual enterprises and cannot study the strategic interactions among supply chain member enterprises and banks. Therefore, this paper uses the game theory to study the cooperative behavior between film and television enterprises and the platform, as well as the lending behavior of banks. This can break through the limitations of the one-way causal relationship in traditional decision-making theories and provide a quantifiable framework for the dynamic evolution of complex systems.”

2. The contribution of this article is unclear. It looks like that the authors just wanted to develop a new application of the game theory. The authors can define the contribution in the end of the Introduction section.

After complementing the above two questions, the review can end.

Response: Thank you very much for the valuable suggestion. In the revised version of this paper, we have added the contribution in the end of the Introduction section. The specific description is as follows:

“The contribution of this paper lies in revealing the influence mechanism of the cooperation between film and television enterprises and sales platforms on the lending behavior of banks, as well as the effects of factors such as negotiation ability, the cost of copyright acquisition and the proportion of equity in joint production on dynamic evolution, providing suggestions for the selection of the optimal cooperation model for film and television enterprises.”

---

## [Decision Letter · Decision Letter 2]

25 Jun 2025

Evolutionary game analysis of supply chain financing for film and television enterprises considering co-production

PLOS ONE

Dear Dr.  Zhen,

Thank you for submitting your manuscript to PLOS ONE. After careful consideration, we feel that it has merit but does not fully meet PLOS ONE’s publication criteria as it currently stands. Therefore, we invite you to submit a revised version of the manuscript that addresses the points raised during the review process.

We look forward to receiving your revised manuscript.

Kind regards,

Godwin Banafo Akrong, Ph.D.

Academic Editor

PLOS ONE

Journal Requirements:

Additional Editor Comments:

I encourage you to address all the concerns raised by Reviewer #3 and make the necessary revisions. I look forward to reviewing your revised manuscript.

Reviewers' comments:

Reviewer's Responses to Questions

**Comments to the Author**

Reviewer #2: All comments have been addressed

Reviewer #3: All comments have been addressed

2. Is the manuscript technically sound, and do the data support the conclusions?

Reviewer #2: Yes

Reviewer #3: Yes

3. Has the statistical analysis been performed appropriately and rigorously?

Reviewer #2: Yes

Reviewer #3: Yes

4. Have the authors made all data underlying the findings in their manuscript fully available?

Reviewer #2: Yes

Reviewer #3: Yes

5. Is the manuscript presented in an intelligible fashion and written in standard English?

Reviewer #2: Yes

Reviewer #3: Yes

Reviewer #2: It is evident that the authors have refined and augmented the full paper in response to the feedback provided by the reviewers, and it is my conviction that the revised manuscript fulfils the criteria for publication.

Reviewer #3: I have only one comment remaining. This comment is optional. The review can end after considering this opinion. The specialty of this article is the development of an evolutionary game model. As a reader of this article, I would like to understand the advantage of adopting an evolutionary game model in decision-making.

**Do you want your identity to be public for this peer review?** For information about this choice, including consent withdrawal, please see our Privacy Policy

Reviewer #2: No

Reviewer #3: **Yes: ** Guang Yi Sheu

---

## [Author Response · Author response to Decision Letter 3]

1 Oct 2025

We acknowledge the reviewers’ comments which are valuable in improving the quality of our manuscript. We have revised our manuscript addressing these comments or suggestions. The changes in the revised version are colored blue.

Reviewer 2:

It is evident that the authors have refined and augmented the full paper in response to the feedback provided by the reviewers, and it is my conviction that the revised manuscript fulfils the criteria for publication.

Response: Thank you very much for your valuable suggestions mentioned earlier. Thank you very much for the acceptance of our paper.

Reviewer 3:

I have only one comment remaining. This comment is optional. The review can end after considering this opinion. The specialty of this article is the development of an evolutionary game model. As a reader of this article, I would like to understand the advantage of adopting an evolutionary game model in decision-making.

Response: Thank you very much for the valuable suggestion. In the revised version of this paper, we have added the explanations regarding the advantages of evolutionary game model in two places.

The first place is at the end of the “Introduction”:

“We use the evolutionary game model to reveal the learning behaviors of irrational decision-makers and the dynamic characteristics of the equilibrium strategies.”

The second place is at the end of the “Literature review”:

“Furthermore, the main advantage of the evolutionary game model lies in its dynamic adaptability. It simulates the real process in which decision-makers adjust their strategies through learning.”

---

## [Decision Letter · Decision Letter 3]

22 Oct 2025

Evolutionary game analysis of supply chain financing for film and television enterprises considering co-production

PONE-D-24-18842R3

Dear Dr. Zhen,

We’re pleased to inform you that your manuscript has been judged scientifically suitable for publication and will be formally accepted for publication once it meets all outstanding technical requirements.

Kind regards,

Godwin Banafo Akrong, Ph.D.

Academic Editor

PLOS ONE

Additional Editor Comments (optional):

Reviewers' comments:

Reviewer's Responses to Questions

**Comments to the Author**

Reviewer #3: All comments have been addressed

2. Is the manuscript technically sound, and do the data support the conclusions?

Reviewer #3: Yes

3. Has the statistical analysis been performed appropriately and rigorously?

Reviewer #3: N/A

4. Have the authors made all data underlying the findings in their manuscript fully available?

Reviewer #3: Yes

5. Is the manuscript presented in an intelligible fashion and written in standard English?

Reviewer #3: Yes

Reviewer #3: The authors have revised this article according to all the reviewers' comment. I think this article can be accepted.

**Do you want your identity to be public for this peer review?** For information about this choice, including consent withdrawal, please see our Privacy Policy

Reviewer #3: **Yes: ** Guang Yih Sheu

---

## [Editor Report · Acceptance letter]

PONE-D-24-18842R3

PLOS ONE

Dear Dr. Zhen,

I'm pleased to inform you that your manuscript has been deemed suitable for publication in PLOS ONE. Congratulations! Your manuscript is now being handed over to our production team.

Kind regards,

on behalf of

Dr. Godwin Banafo Akrong

Academic Editor

PLOS ONE